# The Supply–Demand Budgets of Ecosystem Service Response to Urbanization: Insights from Urban–Rural Gradient and Major Function-Oriented Areas

**Zuzheng Li [1], Baoan Hu [1] and Yufei Ren [2,3,*]**

1  School of Ecology and Nature Conservation, Beijing Forestry University, Beijing 100083, China
2  School of Soil and Water Conservation, Beijing Forestry University, Beijing 100083, China
3  State Key Joint Laboratory of Environmental Simulation and Pollution Control, School of Environment, Beijing Normal University, Beijing 100875, China
*  Correspondence: renyf@bjfu.edu.cn

**Abstract:** The differentiation in the urbanization level's impact on the supply–demand budgets of ecosystem services (ESs) from the perspective of the major function-oriented areas is of great significance for formulating sustainable development strategies at the regional level. This study first constructed the research framework of the supply, demand, and supply–demand ratios (ESDRs) of ESs responding to urbanization from the perspective of major function-oriented zoning, and then took the rapidly urbanized Beijing–Tianjin–Hebei Urban Agglomeration (BTHUA) of China as a case from 2000 to 2020. The relationships between three urbanization indicators, gross domestic production (GDP), population density (PD), and artificial land proportion (ALP), as well as ESDRs of ESs were investigated using Pearson Correlation analysis across three major functional areas. The sensitivity of ESDRs to urbanization was further evaluated using the Random Forest model. The results showed that the supply of carbon fixation, water provision, and food provision increased, whereas their demands far exceeded their supplies, resulting in an increased imbalance between ES supply and demand. With the exception of soil conservation, significantly negative relationships were observed between urbanization indicators and the other three ES supply–demand budgets. The ESDRs of water provision, carbon fixation, and food provision were the most sensitive variables that depended on the population density (PD) in almost all functional areas, whereas the ESDR of carbon fixation exhibited the highest sensitivity to GDP in developed urban areas and rural areas within the preferred development area (PDA) and key development area (KDA). This study could provide comprehensive information for decision making and ES management in different functional areas.

**Keywords:** ESDR; urbanization indicators; BTHUA; random forest model; major function-oriented area

## 1. Introduction

Over the past few decades, it has become increasingly evident that human activities are disturbing natural ecosystems [1–3]. For human societies to achieve sustainability, it is essential to successfully understand and manage the relationship between human beings and natural ecosystems. Through mutual connections and interactions, humans and natural ecosystems form an integrated social–ecological system [4]. Ecosystem services (ES), as the link between humans and natural ecosystems, have gained increased recognition and are being used extensively [5–7]. ES supply is defined as the products and services provided by ecosystems to human society, whereas ES demand can be described as the consumption of the products and services by stakeholders in a particular area within a given time [8,9]. Together, ES supply and demand constitute the dynamic process by which ESs flow between natural ecosystems and social systems [9,10]. Therefore, assessments of ESs that incorporate both supply and demand can accurately reflect natural ecosystem

carrying capacities and human disturbance impacts, which are used as a scientific basis for ecosystem management and resource allocation.

Urbanization, one of this century's most important megatrends, affects the social system and natural ecosystem, resulting in a wide range of ecological issues contributing to global environmental change [11,12]. The ecological issues are frequently aggravated by growing population, economy, and artificial land in urban agglomeration, which place pressure on natural ecosystems and lead to high demand for ESs [12,13]. Due to the contradiction between the increase in urban land and population and the degradation of ESs caused by the encroachment of natural ecosystems, the relationship between supply and demand of ESs is more unbalanced [14,15]. Specifically, on the one hand, urbanization accelerates the transformation of natural and semi-natural lands into artificial lands, thus the ESs are limited through the alteration of ecological processes such as the flow of materials, energy, and biogeochemical cycle [16,17]. For instance, [18] analyzed the relationship between ES supply and urbanization in the Pearl River Delta (PRD), and found that regulating and supporting services decreased, while provisioning services increased due to urban expansion. On the other hand, the growing population and economy in urban areas tend to increase resource consumption and pollutant discharge, thus increasing the demand for ESs [19]. For instance, [20] found an increased demand for carbon storage services due to the increase in energy consumption in the process of urbanization. Therefore, sustainable regional development requires a better understanding of how urbanization impacts the supply and demand of ESs.

Most previous studies examined the relationship between urbanization and the supply of ESs [12,19]. Essentially, these studies described dynamics of ES supply caused by land use changes in the process of urbanization, and the multiple dimensions of urbanization such as economy, populations, and built-up land expansion were less comprehensively considered. In recent years, a few studies explored the changes in supply and demand of ESs under the process of urbanization and the spatial matching relationship [18,20]. Moreover, several studies considered urban agglomeration as a multilevel network system that has substantial gradient differences and heterogeneity [13,19]. For instance, [13] explored the differences in the impact of urbanization indicators on ES supply–demand budgets across regions characterized by different urbanization levels. Currently, urbanization partition only considers socioeconomic factors [19], whereas the major function-oriented areas are based on national spatial governance, considering the natural ecological system and socioeconomic system. Moreover, it has not yet appeared to bring spatial governance into the framework of ecosystem services research, especially from the perspective of major function-oriented areas, to explore the impact of urbanization on the supply and demand of ESs in different major function-oriented areas. Therefore, it is necessary to measure the urbanization level and comprehensively consider the impact of urbanization on ES supply and demand in the major function-oriented areas characterized by different urbanization levels.

The Beijing–Tianjin–Hebei urban agglomeration (BTHUA) experienced rapid urbanization, which has brought several environmental problems and further led to ES supply and demand imbalances [12,21]. Since the State Council promulgated and implemented the "National Main Functional Area Planning" in 2011, the cities of Beijing and Tianjin, and Hebei Province, have implemented the major function-oriented areas planning, respectively, aiming at realizing the sustainable development of the BTHUA. However, there are obvious differences in land space development intensity, functional orientation, and development direction among different main functional areas, which may aggravate the imbalance between social and economic development and ecological environment among different major function-oriented areas. Therefore, the BTHUA was chosen as a case in this study for evaluating the impact of three urbanization indicators, i.e., gross domestic production (GDP), population density (PD), and artificial land proportion (ALP), on ES supply–demand budgets. In detail, this study tries to discuss the following issues: (1) How did the ES supply–demand budgets of the BTHUA change during 2000–2020? (2) Across

different functional areas with differing levels of urbanization, how do different urbanization indicators impact ES supply–demand budgets? Analyzing the urbanization affecting the ES supply–demand budgets from the perspective of major function-oriented zoning is of great practical significance for realizing regional sustainable development.

## 2. Materials and Methods

This study proposed a framework to assess the effects of urbanization on ESDRs and support ES management for each functional area in the Beijing–Tianjin–Hebei urban agglomeration (BTHUA) (Figure 1). Firstly, natural and socioeconomic data were collected, and urbanization of population, economy, and land were measured. Secondly, the spatial variations of urbanization indicators (i.e., GDP, PD, and ALP) and four ES supply–demand ratios (ESDRs) were evaluated from 2000 to 2020. Thirdly, relationships between ESDRs and urbanization indicators were analyzed by using Pearson Correlation and the Random Forest model. Finally, several policy recommendations and measures of functional areas were put forward to improve the coordination development between ESs and urbanization.

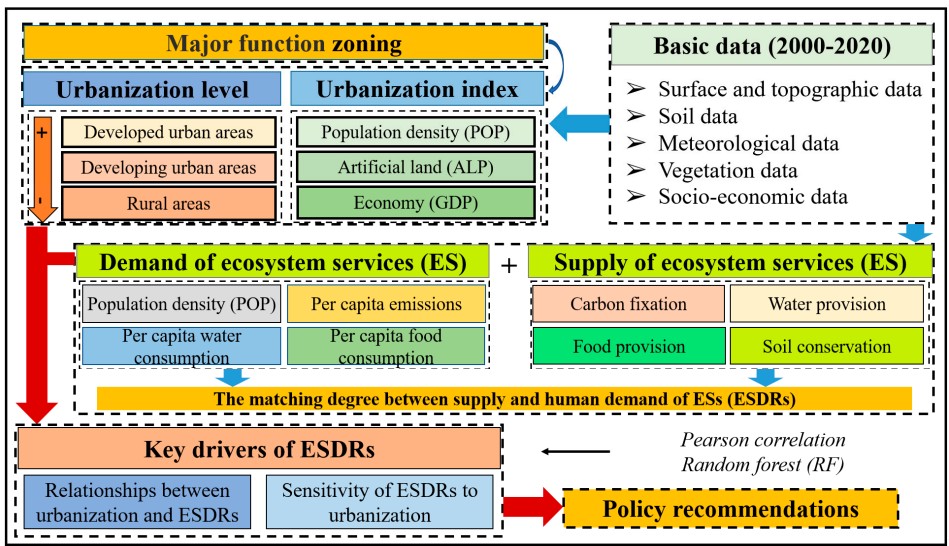

**Figure 1.** The framework and procedures of this study.

### 2.1. Study Area

The Beijing–Tianjin–Hebei urban agglomeration (BTHUA) consists of Beijing, Tianjin, and Hebei Province, and is located in Northern China (35°03′–42°40′N, 113°27′–119°50′E, Figure 2). It covers a total land area of 212,962 km$^2$, accounting for 2.2% of the total land area of China. The northwestern part of this region has a higher elevation and is mostly hilly, while the southeast is relatively flat with an elevation of less than 100 m. This region has a temperate continental climate, with annual mean precipitation of 420–550 mm and an annual mean temperature of −3.5 °C to 24.5 °C [22]. The southeastern plain is an important grain production area in China, growing corn, wheat, and peanuts.

The BTHUA is a political, cultural, and economic center in China, with 13 cities and 173 counties. In 2019, the BTHUA created a GDP of CNY 8.46 trillion, accounting for 8.5% of the country's GDP [23–25]. From 2000 to 2019, the urbanization of this region developed rapidly. The urbanization rate increased from 38.50% in 2000 to 66.70% in 2019. The urban population of the BTHUA increased from 70.91 million to 113.07 million. With its high-density population and high rate of urbanization, the region has experienced long-term water resource shortages and unbalanced development between the sown area and food yield [26]. Moreover, the imbalance between resource supply and demand has triggered a series of regional problems, including surface runoff decrease, land desertification, ground-water over-extraction, air quality deterioration, biodiversity reduction, and ecosystem

degradation [12,17]. Therefore, a comprehensive diagnosis of ESs from the supply–demand perspective is highly significant for the BTHUA's sustainable ES management.

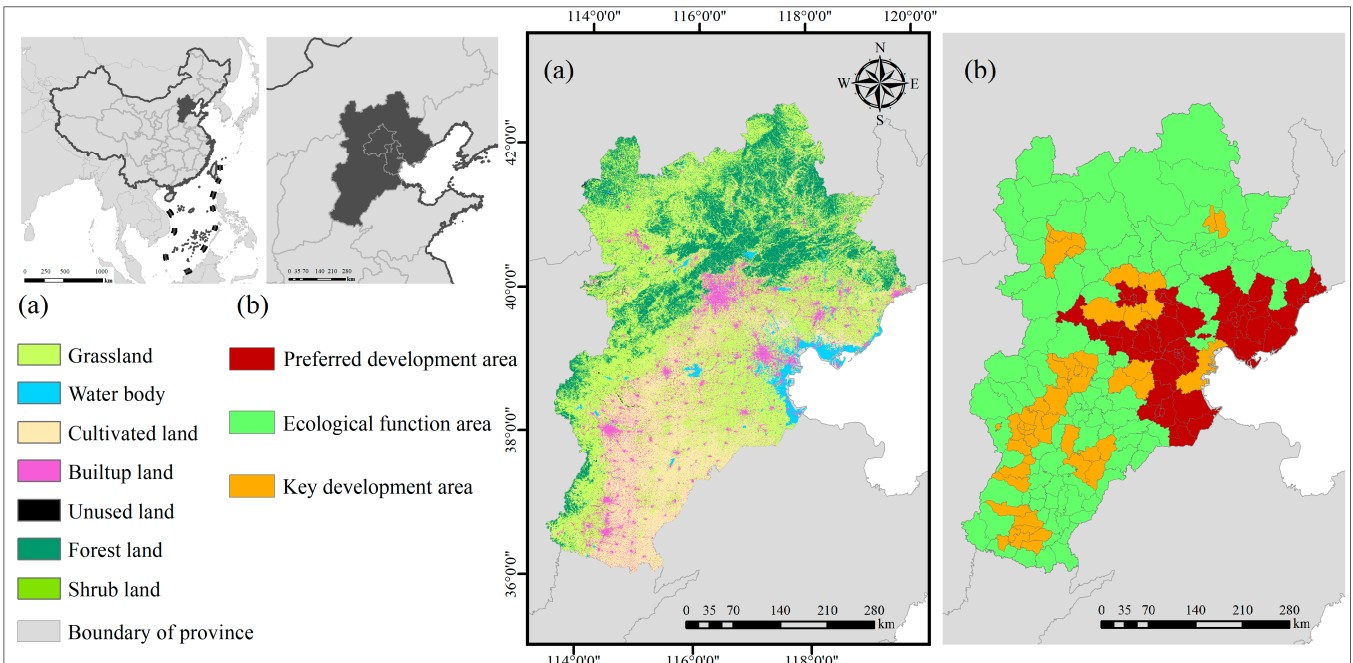

**Figure 2.** Study area. (**a**) LULC distribution of cover types. (**b**) Map of major function-oriented areas, including preferred development area (PDA), ecological function area (EFA), and key development area (KDA).

### 2.2. Data Sources and Processing

This study used eight sets of geospatial and statistical data, including land use, socioeconomic, Digital Elevation Model (DEM), meteorological, energy consumption, soil properties, watersheds, and Net Primary Productivity (NPP). The sources and descriptions of each data used in this study are presented in Table 1. All vector and raster data were converted to the same projection coordinate system (Bei-jing_1954_3_Degree_GK_CM_114E), and the spatial accuracy of all raster data was modified to 1000 m by resampling in ArcGIS 10.3.

**Table 1.** The data requirements, sources, and descriptions.

| Data Types | Data Sources | Sources and Descriptions |
|---|---|---|
| Land use | CASEarth (https://data.casearth.cn/, accessed on 2 March 2022) | Raster (30 × 30 m). Land use types in 2000, 2010, and 2020 were divided into six categories: grassland, water body, cultivated land, artificial surface, unused land, forest land, and shrub land [27]. |
| Socioeconomic data | WordPop (https://www.worldpop.org/, accessed on 4 March 2022) | Raster (1000 × 1000 m). Including population density (PD) and gross domestic product (GDP). |
| Digital Elevation Model data (DEM) | Geospatial Data Cloud (http://www.gscloud.cn/, accessed on 6 March 2022) | Raster (30 × 30 m). A value of elevation for each grid cell. |
| Meteorological data | NCAR (https://climatedataguide.ucar.edu/, accessed on 8 March 2022) | Raster (1000 × 1000 m). Including annual average precipitation and evapotranspiration. |
| Statistical data | Economic Yearbook of Beijing, Hebei, and Tianjin, and the *China Energy Statistical Yearbook* | Excel format. Per capita annual energy consumption and water use data. |
| Soil properties | Harmonized World Soil Database (HWSD; https://www.fao.org/soils-portal/data-hub/soil-properties/en/, accessed on 8 March 2022) | Raster (1000 × 1000 m). Including root restricting layer depth, plant available water content (PAWC) range, etc. |
| Watersheds | HydroSHEDS (https://hydrosheds.org/, accessed on 8 March 2022) | Vector format. A number is assigned for each watershed. |
| Net Primary Productivity (NPP) | USGS (https://www.usgs.gov/, accessed on 8 March 2022) | Raster (500 × 500 m). The amount of organic matter accumulated by plants in unit area and time. |

*2.3. Quantification of Ecosystem Services Supply and Demand*

In this work, four key ESs were selected according to the latest version of the Common International Classification of ES (CICES) (https://cices.eu/resources/ (accessed on 6 March 2021)). The selected ESs are very important for the sustainable development of the BTHUA and are sensitive to global climate changes, land use changes, and human activities [12,22,28]. Table 2 provides an overview of the ESs evaluated in the study area and the reasons for their selections by literature reviews.

**Table 2.** Ecosystem services evaluated in the Beijing–Tianjin–Hebei urban agglomeration.

| Ecosystem Services | Selection Reasons |
| --- | --- |
| Carbon storage (CS) | The absorption of $CO_2$ by vegetation is of great significance to regional climate change and directly affects human health. |
| Water provision (WP) | Water resources are recharged by terrestrial and aquatic ecosystems, affecting the growth of vegetation. |
| Food provision (FP) | Food production is mainly provided by cultivated land and is the basic material for human survival. |
| Soil conservation (SC) | Reduction in soil erosion caused by storm runoff and topography is important in the BTHUA. |

2.3.1. Carbon Fixation

(1) Supply

Carbon fixation (CF) refers to the capacity of vegetation to store carbon dioxide in the atmosphere through photosynthesis, which is essential for climate change mitigation [29]. Here, we measured CF's supply capacity by using the amount of carbon dioxide consumed in photosynthesis [30]. According to the photosynthesis formula, producing 1 g dry matter needs 1.63 g carbon dioxide, then the supply of CF was calculated as follows:

$$S_{CS} = 1.63 \times NPP_i \tag{1}$$

where $NPP_i$ is the value of $NPP$ for pixel $i$.

(2) Demand

The total amount of carbon emissions was used as the demand for carbon fixation in this study, as the changes in per capita emissions will lead to an increase or decrease in the demand for carbon fixation [31]. The total amount of carbon emission is calculated by multiplying the total energy consumption by the standard carbon emission coefficient according to the data provided by $CO_2$ emission inventory of the BTHUA (e.g., residential, industry, and agriculture) [23–25]. The amount of carbon contained in $CO_2$ was estimated to be about 27% of total carbon. We multiplied the average value of emissions per capita by the population density to obtain the total carbon emitted per pixel [32]. The calculation of the total amount of carbon emissions is as follows:

$$D_{CP} = D_{pcfc} \times P_{pop} \tag{2}$$

where $D_{CP}$ is the demand for carbon fixation (t), $D_{pcfc}$ is the per capita carbon emissions (t), and $P_{pop}$ is the population density (person·km$^2$).

2.3.2. Water Provision

(1) Supply

Water provision (WP) refers to the annual quantity of water yield available to humans within a given region [33]. The "Hydropower Water Yield module" of InVEST was used to quantify water provision based on the Budyko curve, with the data including average annual precipitation, root restricting layer depth (mm), plant available water content,

annual reference evapotranspiration (mm), and land use maps [34]. The calculations of annual water provision $Y$ for each pixel are as follows:

$$Y = (1 - AET/P) \times P \tag{3}$$

$$AET/P = (1 + PET/P) - [1 + (PET/P)^{\omega}]^{1/\omega} \tag{4}$$

$$PET = K \times ET_0/P \tag{5}$$

$$\omega = Z \times AWC/P + 1.25 \tag{6}$$

$$AWC = Min\ (Rest.\ layer.\ Soil\ Depth,\ Root.Depth) \times PAWC \tag{7}$$

where $AET$ is the annual actual evapotranspiration (mm); $P$ is the annual precipitation (mm); $AET/P$ is based on an expression of the Budyko curve proposed by [35,36]; $PET$ is the potential evapotranspiration; $\omega$ is an empirical parameter that characterizes the natural climatic–soil properties; $ET_0$ is the reference evapotranspiration; $K$ is the vegetation evapotranspiration coefficient associated with the land use [34]; $AWC$ is the volumetric plant available water content; $Z$ is the empirical constant, and sometimes referred to as "seasonality factor" (1–30); and $PAWC$ is the plant available water content fraction (0–1) [34].

(2)　Demand

　　Water demand ($D_{WY}$) refers to the total amount of water consumption for agricultural and industrial production, inhabitants, and ecological purposes [37,38]. For this estimate, we collected the spatially explicit population density data and the water resource bulletins from each county which provide the water consumption per inhabitant per studied year [23–25]. The calculation of water demand $D_{WY}$ is as follows:

$$D_{WY} = D_{pcwc} \times P_{pop} \tag{8}$$

where $D_{pcwc}$ is the per capita water consumption and $P_{pop}$ is the population density (person·km$^2$). For details, please refer to Table S1.

### 2.3.3. Food Provision

　　Both the supply and demand for food provision (FP) were estimated through statistical data. Here, for FP supply, we first added up each county's production of grain, vegetables, and fruit products, which were the three main types of foods produced in cropland (https://data.cnki.net/Yearbook/Navi?type=type&code=A, accessed on 8 April 2022). Then, we estimated the demand for FP by multiplying the per capita food consumption by the population density. According to [39], the per capita food demand was 322.07 kg·a$^{-1}$, which can basically meet China's per capita food security. For counties without per capita food consumption data, city or provincial-scale per capita food consumption was used as an alternative. Food provision supply and demand can be calculated using the following equations:

$$S_{FPi} = P_{(i,j)} \tag{9}$$

$$D_{FPi} = D_{pcfp} \times P_{pop} \tag{10}$$

where $S_{FPi}$ is the FP supply for county $i$; $P_{(i,j)}$ is the annual provision of $j$ type food for each county, including grains, vegetables, and fruits; $D_{FPi}$ is the FP demand for county $i$; $D_{pcfp}$ is the per capita food demand; and $P_{pop}$ is the population density (person·km$^2$).

### 2.3.4. Soil Conservation

(1)　Supply

　　The "Sediment Delivery Ratio module" of InVEST was used to map and calculate the total amount of soil conservation per pixel based on the universal soil loss equation (*RUSLE*) [34]. Soil conservation equals the difference between the actual amount of soil

erosion (*USLE*) and the maximum potential amount of soil erosion (*RKLS*), assuming the original land cover without the C or P factors [40,41], and is calculated as:

$$SC = RKLS - usle \tag{11}$$

$$usle = R \times K \times LS \times C \times P \tag{12}$$

$$RKLS = R \times K \times LS \tag{13}$$

where *SC* is the amount of soil conservation (tons·ha$^{-1}$·yr$^{-1}$); *usle* is the amount of actual soil loss (tons·ha$^{-1}$·yr$^{-1}$); *RKLS* is the amount of potential soil loss (*tons·ha$^{-1}$·yr$^{-1}$*); *R* is rainfall erosivity (MJ·mm (ha $\times$ hr)$^{-1}$); *K* is the soil erodibility factor for each pixel (MJ·mm (ha $\times$ hr)$^{-1}$); *LS* is the slope length-gradient factor (dimensionless); and *C* and *P* are the crop management and support practice factors for each pixel (dimensionless), respectively.

(2)    Demand

The amount of actual soil loss was used to define the demand for soil conservation, which is based on the quantity of actual soil erosion that human beings are expected to deal with [42]. The calculation of the amount of actual soil loss is below:

$$usle = R \times K \times LS \times C \times P \tag{14}$$

where *usle* is the amount of actual soil loss (tons·ha$^{-1}$·yr$^{-1}$). For details, please refer to Table S2.

### 2.3.5. Ecosystem Service Supply–Demand Ratio (ESDR)

The ecosystem service supply–demand ratio (*ESDR*) index was used to quantify the relationship between the actual ES supply and human demand, identifying ES supply–demand shortages and mismatches [37]. The ESDR index is calculated as follows:

$$ESDR_i = (S_i - D_i) / (S_{max} + D_{max}) / 2 \tag{15}$$

where $S_i$ and $D_i$ refer to the actual ES supply and demand for pixel *i*, respectively, and $S_{max}$ and $D_{max}$ indicate the maximum value of actual ES supply and human demand in the county, respectively. A value greater than 0 indicates an ES surplus, meaning that supply can meet demand, a value of 0 indicates ES supply–demand balance, and a value lower than 0 indicates a deficit—supply cannot meet demand.

### 2.4. Urbanization Classification

As a long-term and complicated process of social and economic development, urbanization has often been measured from three perspectives, i.e., economic urbanization, population urbanization, and land urbanization [13]. To be specific, gross domestic production (GDP, ten thousand yuan/km$^2$), population density (PD, persons/km$^2$), and artificial land proportion (ALP, %) were used to represent economic urbanization, population urbanization, and land urbanization, respectively [13,19]. Moreover, three sub-regions with varying levels of urbanization were identified based on nighttime light data in 2000 and 2020 (Figure 3a). Specifically, the nighttime light data in 2020 was first equally divided into 8 categories according to the quantile method, and then the same method was applied to the nighttime light data in 2000. The second to eighth categories in 2000 were identified as developed urban areas (E). Areas other than urban areas of the nighttime light data in 2020 were identified as developing urban areas (I), and the last remaining areas were rural areas (R). There was a significant stratification effect among the three regions of urbanization in 2020 (Figure 3b).

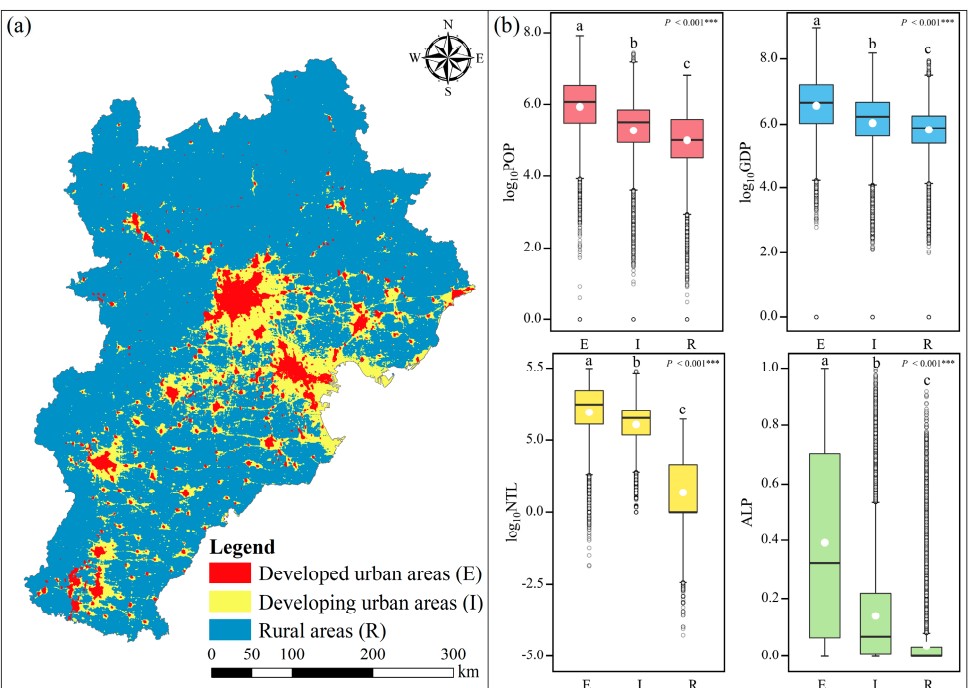

**Figure 3.** Spatial characteristics of different urbanization levels of the BTHUA. (**a**) Distribution of urbanization levels. (**b**) Urbanization indicators, i.e., $\log_{10}$ (PD), $\log_{10}$ (GDP), and $\log_{10}$ (ALP) of E, I, and R. E, I, and R represent developed urban areas, developing urban areas, and rural areas, respectively.

### 2.5. Function Division

According to the major function-oriented area planning issued by Beijing, Tianjin, and Hebei provincial governments, this study divided the BTHUA into key development area (KDA), preferred development area (PDA), and ecological function area (EFA) (Figure 2b) [43]. KDA is defined as a fully urbanized area with the highest development intensity, and its main function is to optimize development. PDA is defined as the area with the greatest development potential and the level of urbanization that needs to be improved, in which the main function is to focus on the development of the construction land. EFA is an important area to ensure regional ecological security and water resource conservation, in which the main function is to restrict development and to restrict large-scale and high-intensity industrialized urbanization development.

### 2.6. Statistical Analysis

To explore how urbanization impacts different ESDRs, Pearson Correlation analysis was applied to assess the relationships between three urbanization indicators and ESDRs across three functional areas. Before the analysis, in order to ensure the unity of all variable dimensions, the variables of PD and GDP were log10 transformed. Random Forest (RF) model is a powerful machine learning tool with high prediction accuracy by using an ensemble of decision trees based on bootstrapped samples from a dataset [44,45]. With the help of this model, multiple collinear problems can be avoided, and independent variables can be assessed separately [46]. Then, the RF model was performed to identify the dominant factors within PD, GDP, and ALP driving ESDRs. Accordingly, the proportion of increased MSE (IncMSE) was used as a measure of the relative importance of each independent variable, in which the IncMSE was calculated as the ratio of the increased mean square error (MSE) to its initial MSE values [47]. A higher IncMSE indicates greater importance of the independent variable. The significance of each predictor was determined using the "rfPermute" package, and the RF model was performed using the "randomForest" package in the R version 4.1.2 (https://cran.r-project.org/, accessed on 3 June 2022).

## 3. Results

### 3.1. Spatiotemporal Dynamics of Mismatch of ES Supply and Demand

The supply, demand, and ESDRs of the carbon fixation, water provision, food provision, and soil conservation were distributed unevenly and had significant changes in the BTHUA from 2000 to 2020 (Figures 4, S1 and S2). The high supply areas of soil conservation, water provision, and carbon fixation service were mainly located in the northwestern parts, whereas the food provision supply was larger in the southeastern parts (Figure S1). The high demand for carbon fixation, water provision, and food provision was mainly distributed in the central area of these cities and increased rapidly (Figure S2). The spatial distribution of supply and demand of water yield, food provision, and carbon fixation showed a serious mismatch, especially in the southeastern parts and central areas of these cities. The soil conservation service has always been in balance in total and in surplus in the mountains, with a slight increase in the ESDR in the southeast parts and a decrease in the other areas. From 2000 to 2020, the ESDRs of carbon fixation, water provision, and food provision were increasing in the city centers of Hebei province and were decreasing in Beijing and Tianjin city centers.

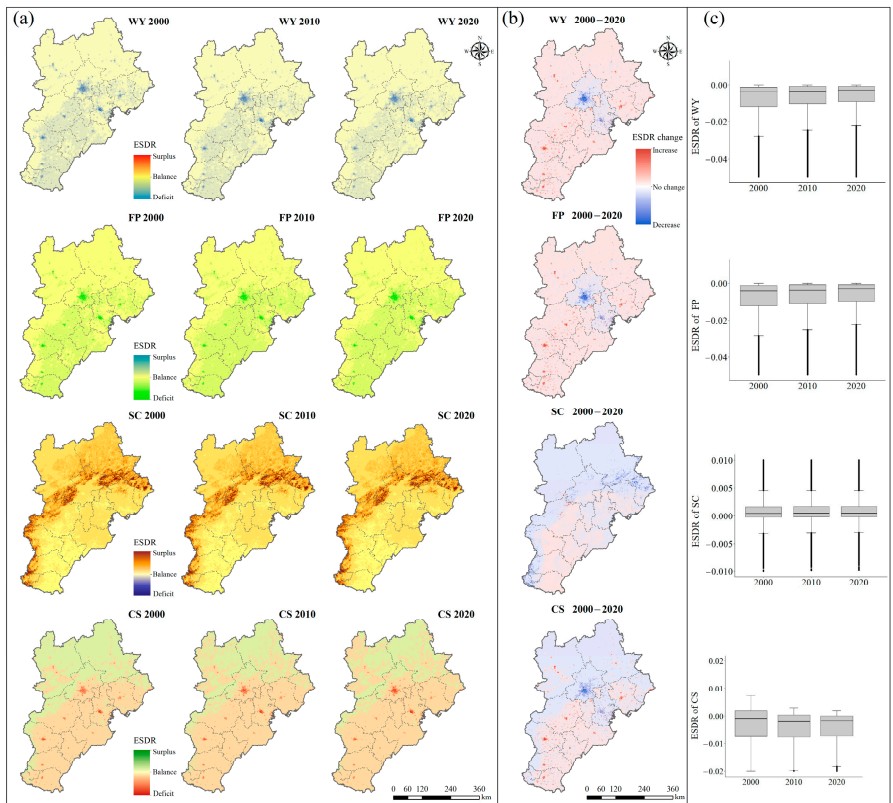

**Figure 4.** Spatial pattern and change of four ESDRs for carbon fixation (CF), water provision (WP), food provision (FP), and soil conservation (SC) from 2000 to 2020. (**a**) Spatial patterns of ESDRs, (**b**) spatial changes of ESDRs, and (**c**) changes in mean values of ESDRs.

The total supply of carbon fixation, water provision, and food provision increased from 53.32 billion tons, 58.43 billion m$^3$, and CNY 113.66 billion in 2000 to 72.29 billion tons, 59.25 billion m$^3$, and CNY 408.10 billion in 2020, respectively (Table S3). The food provision supply has increased substantially, while water production and soil conservation have changed very little. In contrast, the demand for water provision decreased from 35,348.99 billion m$^3$ to 31,065.73 billion m$^3$. The demand for carbon fixation and food provision increased from 242.49 billion tons and CNY 286,845.22 billion in 2000 to 1047.48 billion tons and CNY 1,148,838.66 billion in 2020, with a dramatic increase of 331.96% and 300.51, respectively (Table S4). Hence, although carbon fixation, water provision, and food provi-

sion have been changed to various extents from 2000 to 2020, their demand far exceeded their supply in total (Table S5).

### 3.2. Relationships between Urbanization and ESDRs

In general, urbanization had a significant negative influence on ESDRs of ESs except soil conservation (SC) in all functional areas (Figure 5). The relationships between urbanization indicators and ESDRs were different under different urbanization levels within different functional areas. Population density (PD) has the strongest correlation with ESDRs of water provision (WP), food provision (FP), and soil conservation (SC) in all functional areas. GDP and ALP had the strongest correlation with ESDRs of water provision (WP), food provision (FP), and carbon fixation (CF) in developed urban areas within preferred development (PDA) and key development areas (KDA). In rural areas within ecological function areas, GDP had the strongest correlation with ESDRs of water provision (WP), food provision (FP), and carbon fixation (CF). Moreover, all urbanization indicators had a low correlation with ESDRs of soil conservation (SC).

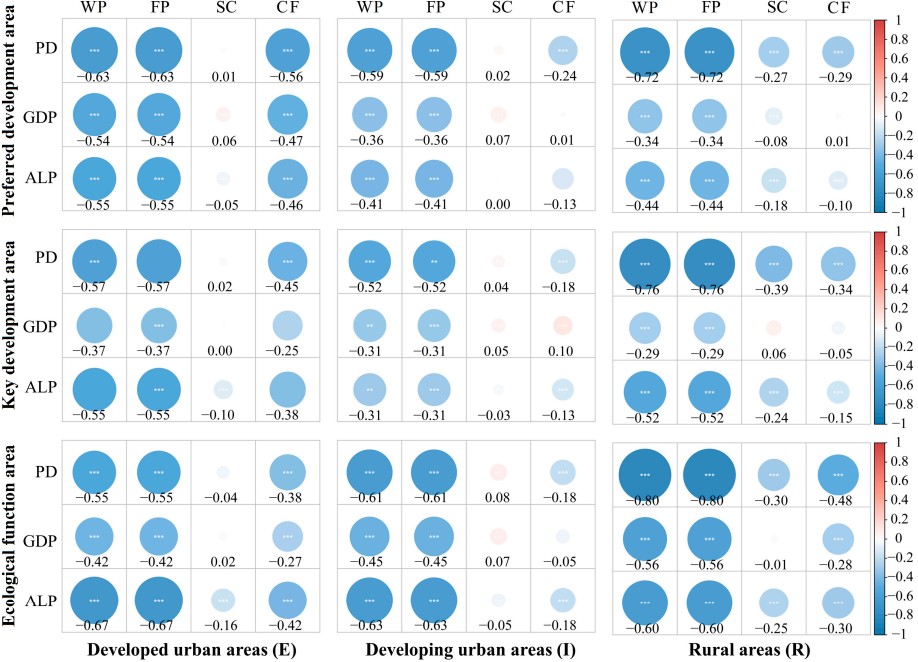

**Figure 5.** Relationships between urbanization and ES supply–demand ratios (ESDRs) in BTHUA in 2020. Three functional areas were distinguished, i.e., key development area (KDA), preferred development area (PDA), and ecological function area (EFA). Four ESs are water provision (WP), food provision (FP), carbon fixation (CF), and soil conservation (SC). Urbanization indicators include population density (PD), gross domestic production (GDP), and artificial land proportion (ALP).

### 3.3. Sensitivity of ESDR to Urbanization

The ESDRs of water provision and food provision services were more sensitive to PD in almost all functional areas, with an increased MSE (IncMSE from 47.08 to 96.72%), followed by GDP and ALP (Figure 6). The results also showed that the ESDR of carbon fixation was most sensitive to PD in developed urban areas and rural areas within the PDA and KDA, with the IncMSE ranging from 33.62% to 89.84%. The ESDR of carbon fixation was most sensitive to GDP in developing urban areas within the PDA and KDA, with IncMSE of 72.18% and 80.35%, respectively. Urbanization indicators had the highest explanation for ESDR of water provision, food provision, and soil conservation services of developed urban areas and developing urban areas within the preferred development areas (variance explained from 6.69% to 98.85%). Urbanization indicators had the highest

explanation for the ESDR of water provision and food provision services of rural areas in ecological function areas (variance explained from 36.81% to 99.21%).

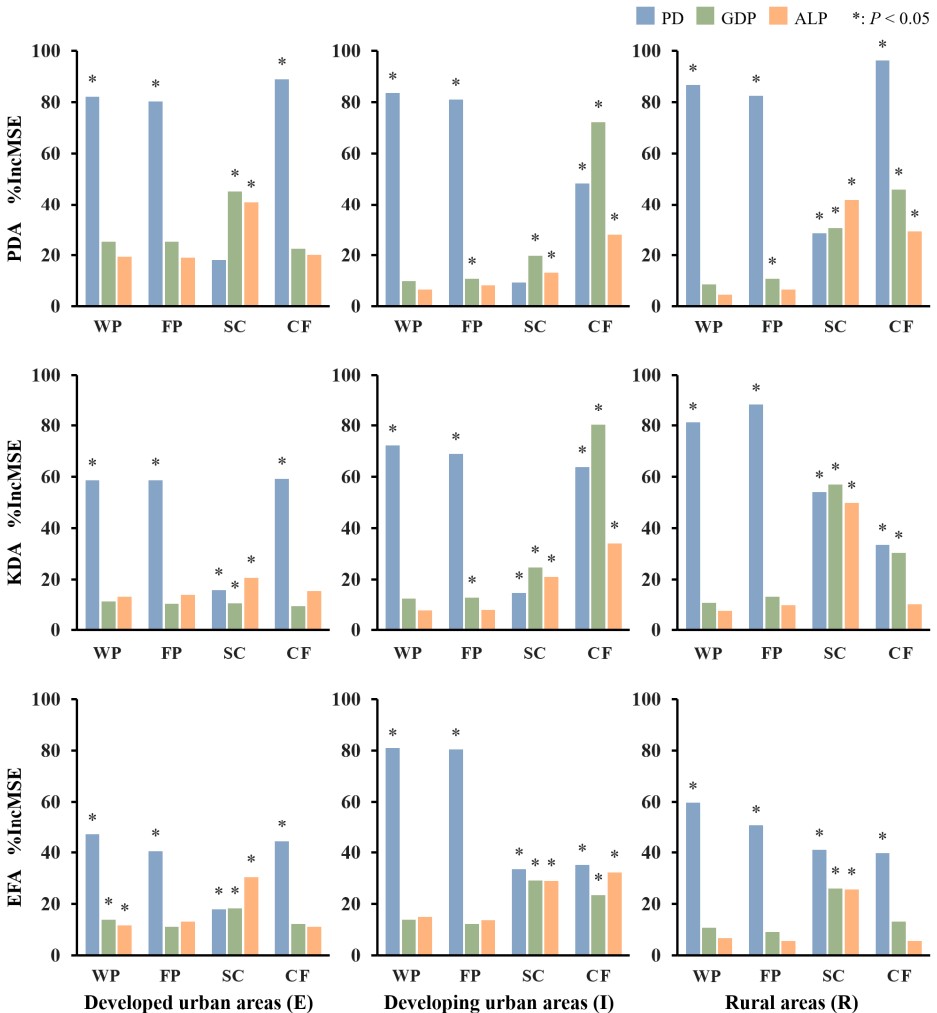

**Figure 6.** Sensitivity of ES supply–demand ratios (ESDRs) for carbon fixation (CF), water provision (WP), food provision (FP), and soil conservation (SC) to urbanization in BTHUA in 2020. Three functional areas were distinguished, i.e., key development area (KDA), preferred development area (PDA), and ecological function area (EFA).

## 4. Discussion

### 4.1. The Impact of Urbanization on ESDRs in Different Functional Areas

As a result of sensitivity analysis, we identified the most sensitive urbanization indicators of ESDRs. ESDRs of water provision, food provision, and carbon fixation were most sensitive to PD, with the exception of developed urban and rural areas within the PDA and KDA. This is attributed to the development of urbanization led to the aggregation of the population in most areas, thus leading to the insufficient ES supply [13,19]. The most important problem appears to be related to the dense population, which has led to a high rise in demand and a steep decline in supply for ESs, creating an imbalanced state between ES supply and demand. Particularly, it was clear that the ESDR of carbon fixation was more sensitive to GDP and ALP in developing urban areas within the PDA and KDA. Since the developing urban areas of PDA and KDA were experiencing rapid urbanization, economic growth and land use changes resulted in a large number of carbon emissions and affected the ESDR of carbon fixation, which is consistent with the research by [19]. The ESDR of soil conservation in rural areas within the KDA and PDA was more sensitive to ALP than in other areas. This is because rural areas have been in the primary stage of urbanization

in the past few decades, which is dominated by the expansion of artificial land; with a growing number of natural and semi-natural lands that have been converted into artificial lands, erosion occurs easily [48,49].

In general, ESDRs were more sensitive to urbanization indicators in developed urban and rural areas within the PDA than in KDA and EFA, which is consistent with the strategic positioning of the main functional areas. On the one hand, the KDA and PDA have a high degree of land space development, which undertake the tasks of absorbing population, economic development, and industrial agglomeration, and are the main pressure areas of water resources, environmental pollution, and social environmental problems [50]. On the other hand, the EFA is responsible for regulating regional climate, conserving water sources, and preventing soil desertification, and it is the type of area with the largest carbon pool [51]. Moreover, the EFA has a wide planting area of grain and vegetables, a large amount of forest land in the field, a high forest coverage rate, and therefore the second largest carbon pool. The ESDR of carbon fixation in developing urban areas within the KDA was more sensitive to urbanization indicators than that of the EFA and PDA. This is mainly because the KDA's main development goal is to promote industrialization and urbanization, while the PDA's goal is to promote energy conservation and emission reduction, strictly control the production of pollutants, and strengthen ecological construction [43].

## 4.2. Suggestions for ES Management and Urban Planning

Several suggestions were proposed to mitigate the imbalance between ES supply and demand, which would contribute to the sustainable development of the BTHUA. The PDA, which is characterized by extremely high urban population density, should first advocate intensive land use and restrict the development of land. For instance, some of the original industrial storage land in developing urban areas has been left idle for the purpose of industrial structure adjustment, and urban planning should be recovered and reallocated [52]. Secondly, the industrial workshop should be three-dimensional, and the plane layout and vertical layout of the workshop should be overlapped to form a complex industrial building. Thirdly, high-rise and small high-rise residential buildings by developers are encouraged [53].

The KDA, as a key development and construction area of urbanization, should first coordinate space development, for instance, strengthening the integration of land and space resources and increasing the space for urban construction in developing urban areas [54]. Secondly, accelerating population agglomeration is necessary, such as expanding the population gathering capacity of developed urban areas and promoting the concentration of rural populations in developing urban areas [55]. Thirdly, protecting the ecological environment is needed, such as coordinating urban and rural environmental protection, strengthening the protection of shelterbelts, increasing urban green space, and protecting and restoring ecological functions [56].

The EFA, as the main supply area of multiple ESs, should first carry out ecological restoration and construction, for instance, strengthening the construction of ecological function zones, effectively restoring and upgrading ecological functions, and improving the production capacity of ecological products [57]. Secondly, it is necessary to maintain the integrity of the ecosystem and improve ecological functions such as water conservation, soil and water conservation, and wind and sand prevention. Thirdly, optimizing the industrial structure is needed, such as strictly controlling the development intensity in developing urban areas, developing the ecological economy, and developing other suitable industries that do not affect the positioning of main functions [58]. Fourthly, orderly guiding of the population transfer to urban areas is needed, such as accelerating the ecological migration and rationally guiding the population in plateau mountainous areas to move to developed and developing urban areas [59,60].

However, the major function-oriented area can play a variety of functions, because the main functions of a certain area and the main contents and tasks of its development do not exclude the area from playing other functions. For instance, as urbanized areas, the major

functions of the PDA and KDA are to provide industrial products and service products and to gather the population and economy. Meanwhile, it is also necessary to protect agricultural space such as basic farmland and ecological space such as forests, grasslands, water surfaces, and wetlands, and to provide a certain amount of agricultural products and ecological products [61]. In addition, the major function of the EFA is to provide agricultural products and ecological products, to ensure the stability of the ecological system, to allow the moderate development of industries that do not affect the major functional orientation, and to allow the necessary urban construction. Therefore, we should carry out orderly development based on the carrying capacity of natural resources and environmental capacity, vigorously develop a green economy and low-carbon economy, and promote the coordination of population distribution and town and economic layout with resources and the environment.

## 5. Conclusions

In this study, the differentiation in urbanization level's impact on the ESDRs was explored across three functional areas in the BTHUA. The results showed that the demands for carbon fixation, water provision, and food provision were growing much faster in urban areas than their supplies from 2000 to 2020, indicating a mismatch of ES supply and demand both in quantity and space. With the exception of the ESDR of soil conservation, urbanization indicators were negatively related to the other three ESDRs. In addition, sensitivity analysis showed that the ESDRs of carbon fixation, water provision, and food provision were most sensitive to population density (PD) in almost all functional areas, whereas carbon fixation also exhibited the highest sensitivity to GDP in developed urban areas and rural areas within the preferred development area (PDA) and key development area (KDA). Five suggestions were put forward for decision making and ES management in different functional areas.

Despite its positive findings, this study does have some limitations that should be highlighted and explored further. First, this study did not comprehensively analyze all types of ecosystem supply and demand, since the coupled system of human and natural factors is incredibly complicated. Second, the supply of ESs, such as water provision and soil conservation, is influenced by land use and biophysical parameters, which are modified according to the actual situation in the BTHUA. Thus, the calculation of ES supply may not be accurate, and the regional differences in ES supply patterns cannot be precisely described. Third, local stakeholders with short-term or long-term interests in ESs were not considered in this study. The future could therefore be devoted to quantifying ES supply and demand from multiple stakeholders. Furthermore, it is imperative to identify who provides these ESs, which areas are entitled to consume them, and whether supplies and consumption are limited to a particular area or extend to other regions. Therefore, ES flow identifications should be incorporated into ES assessments to reveal more details about the interactions between natural and social systems and provide more information on ES management.

**Supplementary Materials:** The following are available online at https://www.mdpi.com/article/10.3390/rs14225670/s1, Table S1: Input data for each LULC class in the InVEST 3.8.0 water yield model. Table S2: Input data for each LULC class in the InVEST 3.8.0 sediment delivery ratio model. Table S3: Total supply of ESs in the BTHUA. Table S4: Total demand of ESs in the BTHUA. Table S5: Total ESDRs of ESs in the BTHUA. Figure S1: Spatial patterns of the supply of carbon fixation (CF), water provision (WP), food provision (FP), and soil conservation (SC) from 2000 to 2020. Figure S2: Spatial patterns of the demand of carbon fixation (CF), water provision (WP), food provision (FP), and soil conservation (SC) from 2000 to 2020.

**Author Contributions:** Conceptualization, Methodology, and Writing—original draft, Z.L.; Formal analysis, B.H.; Validation and Resources, B.H.; Software and Data curation, B.H.; Project administration, Y.R.; Conceptualization, Y.R.; Funding acquisition, Y.R.; Supervision, Y.R. All authors have read and agreed to the published version of the manuscript.

**Funding:** This research was supported by the National Natural Science Foundation of China (Grant No. 42101190) and the China Postdoctoral Science Foundation (Grant No. 2020M680018).

**Acknowledgments:** We would like to thank Yangyi Qin for his assistance with technical support.

**Conflicts of Interest:** The authors declare no conflict of interest.

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
