# Peer review of "The Supply–Demand Budgets of Ecosystem Service Response to Urbanization: Insights from Urban–Rural Gradient and Major Function-Oriented Areas"

_remotesensing, doi:10.3390/rs14225670_

Round 1
Reviewer 2 Report
The article contains all the important elements such as discussion, research limitations etc.
I suggest to write a bit more about the approach used (lines 109-114).
Moreover please explain how the suggestions described in 4.2 should be understood, please provide examples of how: "improve the utilization efficiency of land space, and take the road of intensive development" (line 414-415), "improving the utilization efficiency of industrial and mining construction space "(line 419-420) in this particular case study.
Why is title 2.5 the same as 2.6?
Reviewer 3 Report
My comments:
1. The topic of this paper is interesting and it will contribute in related research field.
2. A section of “Related Works” or “Literature Review” is necessary for this paper.
3. The title of “2.6. Function division” is the same as “2.5. Function division”, Why?
4. The section “Conclusions” must be reinforced more. For example, the more contributions to academic research as well as theoretical implications, research limitations, and suggestions for further research.
Reviewer 4 Report
Comments: The supply-demand budget of ecosystem services response to urbanization: Insights from urban-rural gradient and major function-oriented areas
The paper is pertinent in the way that it takes up the ES supply and demand in relation to urbanization trends. The introduction put forths a nice background to the topic and objectives. The framework used for measuring ESDR is much intuitive and interesting, however, this framework is not proposed rather just a framework of methods/steps of this very work. If it was a proposed framework, it should have been proposed after the study findings. This needs to be made clear. There is some redundancy in relation to study area. There is enough information about the study area in the introduction as well while a separate section on study area would imply removing/trimming the earlier content. The results as they stand are highly intuitive as the method to arrive at has been aptly employed while the data for the study and time covered seems highly realistic. The discussuion section is nicely organized but many of the statements lack valid references and it would improve the quality further if proper citation is integrated into the section.
Minor:
The use of ES in the abstract seems confusing as no full expression given anywhere before its first use.
Round 2
Reviewer 4 Report
Thanks for revising the paper. It is now in a good quality.
Author Response
Response to the Editor and Reviewers’ Comments
To Reviewer #4
Q1: Thanks for revising the paper. It is now in a good quality.
Response to Reviewer comment No. 4: Very thanks for the reviewer’s encouraging remarks and valuable comments. We have also corrected several minor grammatical errors.